# Crocetin Extracted from Saffron Shows Antitumor Effects in Models of Human Glioblastoma

**DOI:** 10.3390/ijms21020423

**Published:** 2020-01-09

**Authors:** Alessandro Colapietro, Andrea Mancini, Flora Vitale, Stefano Martellucci, Adriano Angelucci, Silvia Llorens, Vincenzo Mattei, Giovanni Luca Gravina, Gonzalo Luis Alonso, Claudio Festuccia

**Affiliations:** 1Laboratory of Radiobiology, Department of Biotechnological and Applied Clinical Sciences, University of L’Aquila, 67100 L’Aquila, Italy; mancio_1982@hotmail.com (A.M.); floravitale86@hotmail.it (F.V.); 2Laboratory of Cellular Biology, Department of Biotechnological and Applied Clinical Sciences, University of L’Aquila, 67100 L’Aquila, Italy; s.martellucci@sabinauniversitas.it; 3Biomedicine and Advanced Technologies Rieti Center, Sabina Universitas, 02100 Rieti, Italy; 4Laboratory of General Pathology, Department of Biotechnological and Applied Clinical Sciences, University of L’Aquila, 67100 L’Aquila, Italy; adriano.angelucci@univaq.it; 5School of Agricultural Engineering, University of Castilla-La Mancha, 13071 Ciudad Real, Spain; silvia.llorens@uclm.es (S.L.); gonzalo.alonso@uclm.es (G.L.A.); 6Radiotherapy Unit, Department of Biotechnological and Applied Clinical Sciences, University of L’Aquila, 67100 L’Aquila, Italy; giovanniluca.gravina@univaq.it

**Keywords:** saffron extract, crocetin, glioblastoma in vivo xenograft models

## Abstract

Over recent years, many authors discussed the effects of different natural compounds on glioblastoma (GBM). Due to its capacity to impair survival and progression of different cancer types, saffron extract (SE), named crocetin (CCT), is particularly noteworthy. In this work, we elucidated the antitumor properties of crocetin in glioma in vivo and in vitro models for the first time. The in vitro results showed that the four tumor cell lines observed in this study (U251, U87, U138, and U373), which were treated with increasing doses of crocetin, showed antiproliferative and pro-differentiative effects as demonstrated by a significant reduction in the number of viable cells, deep changes in cell morphology, and the modulation of mesenchymal and neuronal markers. Indeed, crocetin decreased the expression of Cluster of Differentiation CD44, CD90, CXCR4, and OCT3/4 mesenchymal markers, but increased the expression of βIII-Tubulin and neurofilaments (NFH) neuronal linage-related markers. Epigenetic mechanisms may modulate these changes, since Histone Deacetylase, HDAC1 and HDAC3 were downmodulated in U251 and U87 cells, whereas HDAC1 expression was downmodulated in U138 and U373 cells. Western blotting analyses of Fatty Acid Synthase, FASN, and CD44 resulted in effective inhibition of these markers after CCT treatment, which was associated with important activation of the apoptosis program and reduced glioma cell movement and wound repair. The in vivo studies aligned with the results obtained in vitro. Indeed, crocetin was demonstrated to inhibit the growth of U251 and U87 cells that were subcutaneously injected into animal models. In particular, the Tumor To Progression or TTP values and Kaplan–Meier curves indicated that crocetin had more major effects than radiotherapy alone, but similar effects to temozolomide (TMZ). An intra-brain cell inoculation of a small number of luciferase-transfected U251 cells provided a model that was able to recapitulate recurrence after surgical tumor removal. The results obtained from the orthotopic intra-brain model indicated that CCT treatment increased the disease-free survival (DFS) and overall survival (OS) rates, inducing a delay in appearance of a detectable bioluminescent lesion. CCT showed greater efficacy than Radio Therapy (RT) but comparable efficacy to temozolomide in xenograft models. Therefore, we aimed to continue the study of crocetin’s effects in glioma disease, focusing our attention on the radiosensitizing properties of the natural compound and highlighting the ways in which this was realized.

## 1. Introduction

Glioblastoma multiforme (GBM) is a particularly aggressive, and therefore important, brain tumor in humans. The standard of care (SOC) for GBM consists of a combination of surgery and radiochemotherapy. Surgery is the major clinical approach [1,2], however, GBM patients have a very poor life expectancy due to the infiltrative and recurrent nature of this disease [2]. Thus, recent researchers focused their attention on the identification of new therapeutic strategies to attempt to ameliorate the clinical outcome. Human glioblastoma consists of a mixture of differentiated glioma cells and tumor stem-like cells (glioma initiating cells (GICs)), which interact with the tumor microenvironment and play a critical role in the progression and maintenance of the disease [3]. This heterogenic cancer cell population exhibits a complex expression pattern of both differentiated and stem like-related markers such as CD44, a transmembrane glycoprotein that is involved in a series of biological functions and was shown to be overexpressed in GBM and GICs [4]. In particular, the secreted form of CD44 is high in GBM patients and correlates with aggressiveness of the disease [5,6]. CD44 is related to epithelial–mesenchymal transition (EMT) and contributes to the invasive capacity, the formation of metastases, and resistance to therapy [5]. The CD90 glycoprotein is commonly used as a marker of EMT and mesenchymal cells, but it is also identifiable in differentiated GBM cancer cells [7]; the molecular and cellular functions of CD90 are not yet clarified. Regardless, the modulation of CD90 expression dramatically influences cell adhesion, migration, and invasiveness of GBM cells, thereby contributing to the onset of a malignant phenotype. Another important marker of mesenchymal differentiation is octamer-binding protein 3/4 (OCT3/4) [8], which, together with SOX2, Myc, and KLF4, is useful in the identification of pluripotent stem cells [8]. This antigen is strongly expressed in GBM, where it not only maintains cancer stem cells and regulates their capacity for self-renewal but is also involved in tumor progression [9]. CXCR4 is a G-protein-coupled receptor that often appears to be over-expressed in GBM [10], making it an important survival, cell proliferation, and EMT marker. This receptor binds SDF-1 and induces resistance to therapies, therefore maintaining the stemness property. Conversely, different markers are commonly used for the characterization of neuronal differentiation. For example, beta 3 tubulin [11] is a marker of neural lineage (also named TuJ1) and its expression is elevated in normal neuronal cells. Generally, glioma cells exhibit reduced expression of beta 3 tubulin due to their undifferentiated state. However, several studies showed that therapies that are able to induce or increase the expression of this protein [12], and therefore induce a certain grade of differentiation, counteract the proliferation of cancer cells and reduce the aggressiveness of GBM. In addition, neurofilaments are the main components of the intermediate filaments of neurons and maintain the structure of a neural cell. In GBM there are low levels of these filaments, suggesting reduced neural differentiation. Thus, independent studies illustrated that the induction of NF proteins in GBM cells causes a reduction in the progression and malignance of the disease [13]. Finally, fatty acid synthase (FASN) protein, a multifunctional enzyme that plays a central role in lipid synthesis, is particularly noteworthy in regard to glioma. This enzyme is often deregulated in several types of cancer [14] including GBM, in which its overexpression correlates with a more advanced disease stage [15]. FASN is also strongly expressed in patients with glioma stem cells (GSCs) and its inhibition reduces the expression of stem cell markers in favor of differentiation markers in GSCs [16]. FASN inhibitors reduce cancer cell viability and proliferation, thereby inducing cell death via apoptosis both in differentiated cells and GICs [14]. In accordance with this, FASN inhibitors could be a very important therapy approach to induce a certain grade of cancer cell and cancer stem cell differentiation, in order to make tumors more sensitive and increase the efficacy of conventional therapy [14,15,16,17,18]. A series of works indicated that histone deacetylase (HDAC) proteins could be suitable therapeutic targets for GBM treatment due to their capacity to epigenetically modify the expression of genes implicated in tumor progression and resistance to the therapy [19,20]. Indeed, it was recently demonstrated by our research group that the use of HDAC inhibitors led to a reduction in glioma cell proliferation and the induction of apoptosis both in adherent and glioma stem-like cells, as well as those sensitized with RT [21]. Interestingly, various authors illustrated how many natural agents exhibited antitumor action by affecting the role of HDAC proteins and other regulators of gene expression. Over recent years, a series of work introduced the effects of different natural compounds on GBM [22,23]. Among these natural agents, a saffron extract (SE) compound named crocetin (CCT) raised significant interest due to its capacity to impair the survival and progression of different cancer types [24,25]. Therefore, we studied the behavior of crocetin using in vitro and in vivo models of glioblastoma for the first time. In particular, we investigated the antiproliferative effects, the inhibition of cell migration, and the capacity of this compound to induce apoptosis in four GBM cell lines. Following this, we determined whether differentiative effects occurred with crocetin treatment in GBM cells. Moreover, we monitored the in vivo growth of GBM tumors and their inhibition with a bioluminescence assay when crocetin was administered. Our results indicated that saffron extract has an important antitumor property in regard to GBM models.

## 2. Results

### 2.1. Crocetin (CCT) Reduces Proliferation and Induces Morphology Changes in Glioma Cells

In order to evaluate the in vitro antitumor effects of crocetin (CCT) in glioma, we treated U251, U87MG, U373, and U138 cell lines with 250 µM and 500 µM CCT for 24–72 h. We observed a gradual inhibition of GBM cell proliferation and morphological change at 24 and 48 h when crocetin treatment was administered. However, we obtained the best results at 72 h of treatment, at which no cytotoxic effects were observed. The cells were photographed to identify the morphological changes induced by CCT and counted to obtain cell proliferation data at 72 h. The GBM cell lines were shown to be sensitive to CCT due to a dose-dependent reduction in cell proliferation with respect to controls (Figure 1A). In particular, CCT was able to reduce U251 cell growth by 63.7% (*p* = 0.015) at 250 µM and 82.5% (*p* = 0.008) at 500 µM. Reductions of 26% (*p* = 0.057) at 250 µM and 82% (*p* = 0.013) at 500 µM were observed in U87MG cells. Similarly, reductions of 18.3% (*p* = 0.44) at 250 µM and of 83.6% (*p* = 0.00003) at 500 µM were observed in U373 cells and reductions of 53% (*p* = 0.045) at 250 µM and 77.2% (*p* = 0.025) at 500 µM were found in U138 cells. Interestingly, all of the GBM cell lines in our study showed deep morphological changes, including shifting from a short and, in some cases, polygonal (U251 cells) morphology to a more elongated and thin cellular shape. This phenomenon was more evident when the dose of the compound increased (Figure 1B). 

### 2.2. Crocetin Reduces the Levels of Mesenchymal Markers and Induces the Increase of Neuronal Markers in Glioma Cells

Next, we wanted to verify whether the previously mentioned morphological changes were correlated with the modulation of differentiation markers. Therefore, we tested the expression of mesenchymal (CD44, CD90, CXCR4, and OCT3/4) and neuronal (beta 3 tubulin and neurofilament) markers using cytofluorimetric analyses (FACS). Figure 2 shows the FACS histograms resulting from these experiments (Figure 2A), as well as their relative percentage values (Table 1) in untreated cells and after 72 h of treatment with CCT (250 µM and 500 µM). We observed that the mesenchymal markers were significantly reduced by CCT.

Interestingly, the expression of CD44 in U373 cells, starting from 72% of positive cell percentage in the untreated (CTRL) cultures, reached 47.8% with 250 µM CCT and 1.8% with 500 µM CCT. The same results were obtained for CD90 (50.3% in CTRL; 48.6% with 250 µM; 2.2% with 500 µM), OCT3/4 (48.2% in CTRL; 47% with 250 µM; 1.8% with 500 µM), and CXCR4 (50.3% in CTRL, 14.9% with 250 µM; 1.2% with 500 µM). Conversely, we observed a progressive increase in neuronal marker expression, particularly NFH (1.7% in CTRL; 47.3% with 250 µM; 58.6% with 500 µM) and beta 3 tubulin (1.9% in CTRL; 42.6% with 250 µM and 50.5% with 500 µM) in U373 cells. Similar results were also obtained for the U87MG, U138, and U251 cell lines with a certain grade of physiological variability for the same markers due to the different genetic characteristics and tumor stages between the cell lines used in the study. For example, CD44 basal levels fluctuated from 42.1% in U251 to 72% in U373, while its expression after CCT treatment in U373 was 1.8% at 500 µM and 30.1% in U138 at 500 µM. Similarly, the basal levels of CXCR4 started from 50.3% in U373 and 74.7% in U87MG to reach 1.2% in U373 at 500 μM CCT or 56.1% in U87 at the same concentration. 

### 2.3. Crocetin Downmodulates HDACs in GBM Cancer Cells

Differentiation events are often associated with epigenetic modifications in which the activity and expression of histone deacetylases play significant roles. Therefore, we checked the expression of some HDACs in our four GBM cell lines that were treated with CCT or left untreated via Western blotting. As seen in Figure 2B, HDAC1 protein levels decreased after CCT administration in all cell lines, a result obtained in a dose-dependent manner. For example, the levels of HDAC1 were reduced by 36% (250 µM CCT) and 81% (500 µM CCT) with respect to untreated cultures (CTRL) in the U251 cell line. Similarly, the levels of HDAC1 were reduced by 68% (250 µM) and 74% (500 µM) with respect to CTRL in U87MG, the levels of HDAC-1 were reduced by 44% (250 μM) and 64% (500 μM) with respect to CTRL in U87MG, and U373 cells exhibited reductions of 59% (250 µM) and 81% (500 µM) with respect to CTRL.

Thus, the expression of HDAC3 was downregulated only in U251 and U87 cells, reaching 37% (250 μM) and 94% (500 μM) in U251 and 27% (250 µM) and 53% (500 µM) in U87MG with respect to the untreated cultures.

### 2.4. CCT Inhibits the Expression of FASN and CD44 Proteins, Inducing Cell Apoptosis and Reducing Migratory Capacity

The expression of FASN and CD44 were observed using Western blotting. The expression of these markers is often related both to resistance to apoptotic events and to the induction of migration capabilities of tumor cells. In Figure 3A, we demonstrated that CCT downmodulated FASN expression levels. The decrease in FASN expression ranged between 16% (U87MG) and 60% (U251) for the 250 µM dose and between 35% (U87MG) and 87% (U138 and U373) at the 500 µM dose when compared to untreated cells. The Western blot analysis revealed that CD44 protein levels were significantly reduced by CCT administration by about 23% (250 µM) and 76% (500 µM) in U87MG cells versus the untreated cultures. The reduction of CD44 was similar to that observed in U373 cells (28% at 250 μM and 77% at 500 μM). The CD44 expression was also reduced in U251 and U138 cells, but to a lower extent, i.e., 7% and 12% at 250 mM and 45% and 47% at 500 mM, respectively. This agreed with the FACS analysis results. Therefore, we verified using the wound-healing repair test that the downmodulation of FASN and CD44 expression resulted in a reduction of the migratory ability of our cell lines in response to CCT. Figure 3B reports the wound repair results related to the U251 and U87 cells, which was only partial after 24 h of testing in the CCT-treated cells. The effects were dose-dependent. 

### 2.5. In Vivo Studies: Crocetin Modifies Tumour Growth of GBM Cells Subcutaneously Injected into Female nu/nu Mice (Subcutaneous Xenograft Model)

The efficacy of CCT as a single therapy was verified in comparison with standard therapeutic approaches (RT, TMZ, or RT + TMZ) in CD1-nu/nu mice with experimental brain tumors using subcutaneous xenograft models of U251 (Figure 4A,B and Table 2) and U87MG (Figure 4C,D, Table 3) cells. A CCT dose of 100 mg/kg/day was chosen according to literature reports. The CCT efficacy was compared with radiotherapy (RT, one dose of 4 Gy), temozolomide (TMZ, 16 mg/kg for 5 consecutive days [9]) and their combination. CCT significantly reduced the tumor progression of the U251 xenografts (Figure 4A,B, Table 2) measured at the end of the experiment (35 days), as well as the percentage of U87MG tumors that progressed (Figure 4C,D, Table 2) over the studied time period, as measured with Kaplan–Meier curves. The evaluation of hazard ratio values (Table 3) demonstrated that, in 87MG xenografts, CCT (HR = 4.69) was more effective when compared to those that were administered 4 Gy of RT (HR = 2.79 vs. vehicle and HR = 3.30 vs. CCT). CCT efficacy was similar to Temozolomide (HR = 4.39 vs. vehicle and HR = 1.67 vs. CCT). In contrast, standard radiochemotherapy, which was based on RT plus temozolomide administration, was significantly more effective with respect to CCT administration (HR = 5.52 vs. vehicle and HR = 2.42 vs. CCT). A comparison of the effects of CCT in U251 xenografts (Figure 4A,B, Table 2) showed that CCT also reduced tumor growth in this xenograft model, thereby increasing the time of tumor progression.

The percentage of tumors that progressed over the time period was also reduced (Table 3), with HR = 4.58 vs. vehicle. TMZ (HR = 4.05 vs. vehicle) was similarly effective to CCT, with an HR = 1.35 for the comparison between TMZ and CCT. The combination treatment with RT and TMZ showed that sole CCT administration was more effective (HR = 5.34 vs. vehicle), with an HR = 2.36 for the comparison between CCT and standard therapy.

Previously, we observed that different experimental GBM xenografts were characterized by the presence of heterogeneous populations of tumor and inflammatory cells [9,24] with giant and multinucleated cells that possessed polygonal or spindle morphologies and abundant, intensely eosinophilic cytoplasm and low-stained nuclei. As seen in Figure 5, the U87MG cells in the control group with a high growth ratio were dispersed on a fibrillar collagen background that enveloped the abundant vasculature (Figure 5A). CCT administration reduced tumor proliferation, thereby producing less compact tumors with a very low expression of the mesenchymal marker CD44 and an increased expression of the neuronal marker β3-tubulin (Figure 5B). 

### 2.6. Crocetin Increased Disease-Free and Overall Survival in Orthotopic Intra-Brain Tumours as Determined by Using Differentiated Luc-U251MG Cell Model

The efficacy of CCT was investigated in mice with experimental brain tumors. We wanted to inoculate a small number of cells (3 × 10^3^) into the brain to simulate post-surgical chemotherapy and radiotherapy treatment. Indeed, in this case, a few tumor cells might remain in the operating bed, which would be able to re-grow and recur. We administered treatment after 5 days when no bioluminescence could be detected intracranially. Next, we treated the animals for 35 days, with a maximum of 200 days follow-up without drug administration (total 240 days). Figure 6A shows the schedule of treatments with the animals having different amounts of bioluminescent signal. Bioluminescence was associated with low, intermediate, and large intra-brain tumors at necroscopy. Figure 6A also shows the signals obtained using the Alliance Mini HD6 system (Uvitec Ltd., Cambridge, UK.), which was issued at 42 days after the intra-brain cell inoculation in 7/9 mice (one animal previously perished) of control and 3/10 of mice treated with CCT.

Parameters pertinent to our analyses were recorded, including: (i) tumor growth delay (the time at which luciferase activity was intracranially detectable), which indicated the recurrence time equivalent to the human parameter DFS; (ii) tumor progression through the analysis of bioluminescence imaging (BLI) photon counts and tumor volumes (calculated by magnetic resonance imaging, MRI); and (iii) the survival time (equivalent to human parameter OS), as indicated above. Panel B of Figure 6 and Table 4 analyze when the luciferase signal equivalent to disease-free survival (DFS) first appeared in mice injected with luciferase-transfected U251 cells. In particular, control mice developed a bioluminescent lesion after 20–50 days with a mean of 29.0 ± 2.53 days and a median of 30.0 days (20.0–35.0 days, 95% CI), whereas, in the CCT-treated animals, the bioluminescence appeared after 20–65 days with a mean of 46.5 ± 4.53 (*p* = 0.0050 vs. the control) and a median of 45.0 days (35–65 days, 95% CI). RT showed a bioluminescent lesion from 20 to 50 days with a mean of 36.0 ± 3.15 (*p* = 0.1174 vs. the control and *p* = 0.0877 vs. CCT) and a median of 32.5 days (30–50 days, 95% CI). The animals treated with TMZ showed recurrent tumors from 30 to 80 days with a mean of 52 ± 5.06 days (*p* = 0.0012 vs. control, *p* = 0.0202 vs. RT and *p* = 0.4523 vs. CCT) and a median of 47.5 (45.0–70.0 days, 95% CI). Standard RT plus TMZ administration significantly delayed the insurgence of recurrent tumors from 50 to 240 days with a mean of 128 ± 25.1 (*p* = 0.0016 vs. control, *p* = 0.0028 vs. RT, *p* = 0.0114 vs. TMZ and *p* = 0.0072 vs. CCT) and a median of 90 days (50.0–160.0 days, 95% CI). Figure 6C shows the overall survival data in days. The mean OS in the control animal group was 71.3 days ± 15.5. CCT delayed this by about 36 days, with a mean of 108.3 ± 23.8 (*p* = 0.0008). CCT was shown to be more effective than RT (88.5 ± 12.0 days, *p* = 0.0324). Temozolomide reached a mean of 117.8 ± 34.9 days, with the results not being statistically different to CCT. The SOC therapy showed a mean of 141.0 ± 38.5. In this case, the efficacy of the combinatory treatment was significantly better than CCT, as indicated in Figure 6C, Table 5. Next, we generated Kaplan–Meier curves (Table 6) and calculated the HRs for each experimental group. The RT treatment showed a nonsignificant increase in terms of DFS, with an HR = 1.67 (*p* = 0.1379) compared to the untreated animals, whereas the CCT (HR = 2.85, *p* = 0.0022), TMZ (HR = 3.38, *p* = 0.0003), and RT + TMZ (HR = 5.21, *p* < 0.0001) treatments significantly reduced the percentage of animals in progression. Although 30% of the CCT-treated animals showed bioluminescence during the course of treatment compared to 70% of the RT-treated animals, the Kaplan–Meier curves indicated that the effects of CCT and RT were not significantly different. Similarly, no statistically significant differences were observed in the comparison between CCT and TMZ, where 10% of the animals showed progression during the pharmacological treatment cycle. By contrast, the combination of RT + TMZ caused no animals to progress during the treatment cycle, which was statistically more active when compared with the other single treatments, including CCT. These results may indicate that (i) the tumor growth rate of each tumor was not different between RT, TMZ, or CCT, and (ii) if the administration time was repeated with further therapy cycles, the differences between these individual administrations could have become significant. 

## 3. Discussion

GBM is an important an aggressive brain tumor brain in humans, characterized by a very poor life expectancy from diagnosis [1] and very low survival rate at two years. Recent research focused on the identification of new therapeutic strategies to improve the outcomes of this neoplasia. In recent years, many authors studied the effects of different natural compounds on GBM [22]; among these, SE raised great interest. Crocetin is the major active product of SE and has an elevated capacity to counteract the survival and progression of different cancer types [24,26,27,28]. However, the effects of crocetin on GBM are poorly understood. In order to explore potential of this compound to treat this kind of tumor in more detail, we reported results regarding in vivo and in vitro models of glioma treated with crocetin. Several authors demonstrated the in vitro antiproliferative action of crocetin in different kinds of tumor cells [29,30,31,32]. In order to evaluate the antiproliferative effect of crocetin on GBM cells, four tumor lines (U251, U87, U138, and U373) were treated with 250 µM and 500 µM of crocetin and counted after 72 h of incubation. We selected these crocetin concentrations based on previous experiments indicating that they were effective. The results showed that all of the cell lines were sensitive to the molecule in a dose-dependent and statistically significant manner, exhibiting a large reduction in proliferation and strong morphological changes. In association with this, deep changes in cancer cell morphology were observed, moving from a short, and, in the case of U251 cells, polygonal morphology to a thinner and elongated cell body shape. Starting with this evidence, to verify whether differentiation occurred, we checked the expression of common mesenchymal and neuronal markers using cytofluorimetric analysis. Surprisingly, the treatment of GBM cell lines with increasing doses of crocetin led to important decreases in CD44, CD90, CXCR4, and OCT3/4, alongside very important increases in βIII-tubulin and NFH.

These results, which were in line with the microscopic considerations, suggested that attempted transitions from mesenchymal to neuronal lineage occurred in the GBM cells exposed to crocetin treatment. Different independent studies showed the capacity of crocetin to induce cancer cell differentiation through the inhibition of particular factors involved in the regulation of gene expression, such as HDACs [33,34]. Indeed, these proteins are important players in the onset and maintenance of tumors due to their inhibitory effects on the expression of suppressor genes in neoplastic disease. This evidence encouraged us to check whether HDAC (histone deacetylase) inhibition occurred in GBM cells treated with crocetin. Western blot analyses revealed reduced expression of HDAC1 and HDAC3 in U251 and U87 cells, but a reduction in HDAC1 expression only in U138 and U373 cells, in a dose-dependent manner. Moreover, although further investigations must be performed, we hypothesize that the epigenetic modifications occurring in GBM cells with crocetin treatment are responsible for the differentiative effects, which may preliminarily explain the mechanism of action of the natural compound. 

FASN is a protein that is frequently highly expressed in glioblastoma, and its expression relates to the malignance of the disease [14,15,16,35,36]. Furthermore, FASN is identified a suitable target for GBM treatment and it was, in fact, demonstrated that its inhibition led to programmed cell death in cancer cells. Previously, we referred to the CD44 protein as a marker of mesenchymal differentiation. However, CD44 in the context of glioma is also associated with adhesion, migration, and cancer cell proliferation, thereby favoring therapy resistance [37,38,39,40]. Considering the relevant meaning of these two markers in cancer progression, we checked the expression levels of FASN and CD44 in U251, U87, U138, and U373 cells which were treated with increasing doses of crocetin. Intriguingly, the results not only showed that effective inhibition of both markers occurred with treatment but an important activation of the apoptosis program was also put in place accordingly, as demonstrated by the appearance of caspase 3 cleaved bands in GBM cells treated with crocetin, which was also described in human acute promyelocytic leukemia cells [32].

Moreover, the wound-healing repair assay, which is a common test used to evaluate cancer cell migration, revealed that crocetin reduced, and actually inhibited, the movement of glioma cells. Thus, we suggest that, consistent with demonstrations in the literature for other tumor cell systems, crocetin administration activates programmed cell death and strongly reduces the migration of glioma cells. These two important antitumor properties may be mediated by the abrogation of FASN and CD44 pro-tumor activity. The in vivo studies aligned with the results obtained in vitro. Indeed, crocetin was shown to inhibit the growth of U251 and U87 cells subcutaneously injected into animal models. In particular, the TTP (time to progression) and the Kaplan–Meier curves indicated that crocetin had more major effects than radiotherapy alone, but similar effects to temozolomide (TMZ). The comparison between crocetin and the combination treatment RT–TMZ (standard of care) indicated that radiochemotherapy was more powerful than crocetin alone. In association with this, immunohistochemistry (IHC) investigations showed, similar to what was seen in vitro, decreased CD44 and increased β3-tubulin expression in the tumor tissues of treated animals. Intra-brain U251 inoculation provided further important indications regarding the effects of crocetin for GBM treatment. We performed the orthotopic model with U251 glioma cells in accordance with previous work indicating that crocetin effectively overcame the blood–brain barrier in animals [41]. In order to simulate the recurrence of the disease, we inoculated a small number of cells and, through the use of luciferase-transfected U251 glioma cells and bioluminescence assays, it was possible to monitor the insurgence and the progression of lesions in animals. The results said that crocetin treatment increased disease-free survival (DFS), inducing a delay in the appearance of detectable lesions with respect to untreated animals and RT-treated animals. However, crocetin showed less efficacy than temozolomide and combination RT–chemotherapy. Crocetin also augmented the mean rate of overall animal survival with respect to the control and RT-treated animals, but was less powerful than temozolomide and the standard of care treatment. These results encouraged us to continue studying crocetin’s effect in glioma disease. In particular, the role of crocetin in combination with ionizing radiation should be investigated, in particular, whether crocetin increases the sensitivity of glioma cells to radiations and the way in which this may be achieved.

## 4. Materials and Methods

### 4.1. Crocetin Extraction

Crocetin was isolated from saffron as previously described [42]. Briefly, we used dry stigmas of pure *Crocus sativus* L. of the protected designation of origin (PDO) “Azafrán de La Mancha”, which guarantees a minimum coloring power of 200 units. Saffron was directly purchased from a producer (Agrícola Técnica de Manipulación y Comercialización en Minaya, Albacete, Spain). Crocetin was obtained by hydrolysis of aqueous solutions of the saffron using a protected internal method by Verdú Cantó Saffron Spain (Novelda, Alicante, Spain). According to HPLC, it had a purity of 96% at 250 nm and 99% at 440 nm; the proportion of trans-crocetin was 86% and cis-crocetin made up 13%. The extraction procedure led to 8–10% of crocetin using the dried stigmas. 

### 4.2. Cell Cultures and Cell Culture Materials

U251, U87MG, U373, and U138 glioma cell lines were purchased from ATCC (Manassas, VA, USA). Culture materials were purchased from Euroclone (Euroclone, Milan, Italy) when not otherwise specified. Cells were cultured in DMEM containing 10% fetal bovine serum (FBS), 1% glutamine, and 1% penicillin/streptomycin at 37 °C with 5% CO_2_ in humidity. 

### 4.3. Western Blot

Untreated and treated cells were processed for protein extraction with lysis buffer 1× (Cell Signaling, Denver, CA, USA), supplemented with phosphatase and protease inhibitor (Sigma, Saint Louis, MO, USA). The whole cell lysate of each sample was subjected to SDS-PAGE and transferred to a nitrocellulose membrane (GE Healthcare, Chicago, IL, USA). The membranes were saturated for 1 h at room temperature with 5% nonfat milk in TBS-Tween and incubated with the following primary antibodies: anti-CD44 (Abcam, Cambridge, UK), FASN (Gene Tex, Zeeland, MI, USA), caspase 3 cleaved (Cell Signaling, Denver, CA, USA), GAPDH (Santacruz, Dallas, TX, USA), HDAC1 (Santacruz, Dallas, TX, USA), and HDAC3 (Santacruz, Dallas, TX, USA) at concentrations recommended by the suppliers. Band identification was performed by exposing the membranes to the enhanced chemiluminescence (ECL) substrate, and the images were acquired using a ChemiDoc Molecular Imager (Biorad, Hercules, CA, USA). The procedures were performed according to the protocols provided by manufacturers.

### 4.4. Fluorescence-Activated Cell Sorter Analysis

Expression of surface antigens in U138, U373, U251, and U87MG cell lines, either treated or untreated with 250 μM or 500 μM crocetin, was quantified using the Fluorescence-Activated Cell Sorter (FACS). Cells were fixed with 4% paraformaldehyde for 10 min at 4 °C. After washing, cells were incubated for 1 h at room temperature with anti-CD44, anti-CD90 (Millipore, Milan, Italy), and CXCR4 (Santa Cruz Dallas, TX, USA). Moreover, we quantified the expression of endogenous antigens using flow cytometry in the same cell lines that were treated as above. The cells were fixed with 4% paraformaldehyde for 10 min at 4 °C. After washing, the cells were permeabilized with 0.1% (v/v) Triton X-100 for 10 min at room temperature. The cells were washed in PBS and incubated for 1 h at RT with anti-NFH, anti-β3-tubulin (Cell Signaling) and OCT3/4 (Santa Cruz). All samples were incubated for an additional 30 min with CY5-conjugated anti-rabbit IgG H&L or PE-conjugated anti-mouse IgG (Abcam Cambridge, UK). Finally, samples were analyzed using a BD Accuri C6 Flow cytometer (Becton Dickinson Italia SpA, Milan, Italy) equipped with a blue laser (488 nm) and a red laser (640 nm). At least 10,000 events were acquired. Negative controls were obtained by analyzing the autofluorescence and samples treated with only the secondary antibodies. Although FACS should be performed on live cells, especially for membrane-associated antigens, we chose to use a fixation in 4% paraformaldehyde to avoid internalization and degradation of the membrane markers.

### 4.5. Wound Healing Repair Assay

Glioma cells were seeded in 3-mm Petri dishes at a density that, after 24 h of incubation, allowed 70–80% confluence to be reached. A scratch was performed with a 200 μL pipette tip on the well of the dish. The detached cells were removed via 2 washes with fresh medium. Treatment with crocetin at the necessary concentration was performed and after 24 h of incubation the differences in wound repair between the untreated and treated cells were checked. The pictures were taken at 0 h and after 24 h to appreciate the level of cell migration for each sample in the study.

### 4.6. Animal Experiments

#### 4.6.1. Subcutaneous Xenograft Model

After 1 week of quarantine, female CD1-nu/nu mice at 6 weeks of age (purchased from Charles River, Milan, Italy) that were followed under the guidelines established by our institution received subcutaneous flank injections (2 each) of 1 × 10^6^ U87MG and U251. After the tumor was established and when it reached 80 mm^3^ in volume, the mice were randomized into the following groups: (a) DMSO/PBS controls (i.p.) injections; (b) temozolomide (TMZ) (PO) 32 mg/kg for 5 consecutive days; (c) CCT (100 mg/kg, per os) in 100 μL of DMSO/PBS; and (d) RT in a single dose of 4 Gy at day 3 from randomization. In order to monitor the toxicity of the treatment, the body weights of the mice were recorded twice a week. Tumor mass growth was evaluated twice a week measuring the diameters of the subcutaneous tumors with a Vernier caliper. Tumor volumes were calculated using the following formula: tumor volume (mm^3^) = 4/3πR1 × R2 × R3 [9,43]. At the final end point of the experiment (35 days after the start of treatment), we sacrificed the animals with carbon dioxide inhalation. The tumor masses were collected, weighed, and fixed in paraformaldehyde for immunohistochemical analysis. 

#### 4.6.2. Evaluation of Treatment Response In Vivo

From the in vivo experiments, we acquired the following series of information: (1) The volume and the weight of the tumors, (2) the tumor progression (TP) corresponding to the time required for the tumor mass to double, and (3) the time to progression (TTP), which was obtained by recording the tumor growth over time. 

#### 4.6.3. Orthotopic Intra-Brain Model

Female CD1-nu/nu mice were inoculated intracerebrally as previously described [9,25] with the luciferase-transfected U251 cell line. Just before treatment initiation (5 days after injection), the animals were randomized into five groups of 10 mice each: (a) DMSO control (i.p.) injections; (b) TMZ per os 32 mg/kg for 5 consecutive days; (c) CCT (i.p.) injections (100 mg/kg) in 100 μL of DMSO/PBS; and (d) RT in a single dose of 4 Gy at day 3 from randomization and (e) SOC (RT + TMZ). In vivo bioluminescence assays were performed using the UVITEC Cambridge Mini HD6 (UVItec Limited, Cambridge, UK). The animals, once anesthetized, were intraperitoneally injected with luciferin (150 mg/kg), subjected to the bioluminescence signal acquisition after 15 min. For the orthotopic experiment, we decided to inoculate a small amount of U251 GBM (3 × 10^3^) cells to simulate recurrence. Indeed, after the administration of radiochemotherapy after surgery, a small number of cells can remain in the wound bed with the potential of inducing a recurrence of the disease. Treatments started 5 days after the cell injection when no positivity to bioluminescence was intracranially detectable. The mice were euthanized when distress signs (e.g., altered gait, tremors/seizures, lethargy) or the loss of 20% of body weight appeared. All procedures were conducted according to animal care laws with approval from the animal welfare department, authorization 555/2017-PR (dated 7 July 2017).

#### 4.6.4. Evaluation of Treatment Response In Vivo

The following parameters were used to quantify the antitumor effects after different treatments as previously described [9,25]: (1) Disease-free survival (DFS), defined as the time necessary to exhibit a positive bioluminescence signal, and (2) overall survival (OS), defined as the time necessary to observe distress signs or animal death. The animals with distress signs were euthanized by CO_2_ inhalation and their brains collected for histological analyses. DFS and OS data were collected and used for statistical evaluation and the generation of Kaplan–Meier curves.

### 4.7. Histochemical and Immunohistochemical Analyses

Paraffin-embedded tissue sections measuring 4 μm were analyzed for Trichromic and indirect immunoperoxidase staining. Immunohistochemical staining was carried out using rabbit anti-CD44 polyclonal antibody (Abcam) and anti-βIII tubulin (Cell Signaling). Briefly, samples were fixed in formalin and embedded in paraffin, subjected to deparaffinization in xylene, and dehydrated in ethanol. The block of endogenous peroxidase was performed at room temperature for 30 min, with 0.1% hydrogen peroxide–methanol. The samples were washed twice with distilled water and incubated with target retrieval solution (Dako, Glostrup, Denmark) in a microwave oven. The samples were the washed twice with distilled water. Non-specific binding was performed with a specific reagent for 15 min. The sections were incubated overnight with anti-CD44 and βIII tubulin antibodies in a dilution of 1:100, at 4 °C. Protein identification was obtained using an Horseradish peroxidase (HRP) conjugated secondary antibody IgG Super Vision Kit (Italian distributor of Boster immunoleader USA, Tema Ricerca, Bologna, Italy) and the Pierce 3,3′-Diaminobenzidine (DAB) Substrate Kit (Thermo Scientific, Italian distributor Tema Ricerca) for 5 min at room temperature. Cells were counterstained using Meier’s hematoxylin. The procedures were performed according to the protocols provided by manufacturers.

### 4.8. Statistics

The software used for statistical analysis and graphic presentation was MedCalc (MedCalc, Ostend, Belgium). We determined the mean, the SD, and the median with 95% CI for our comparison using the ANOVA and Turkey’s test. For multiple comparisons, we corrected the *p*-value according to Bonferroni’s correction. TTP was also analyzed using Kaplan–Meier curves. The logrank test for trends was used for comparisons of multiple survival curves, which examines the probability of a trend in survival rates across the groups. All of the tests were two-sided and determined by Monte Carlo significance. *P*-values of <0.05 were considered statistically significant. 

## 5. Conclusions

Our report shows that crocetin extracted from saffron possesses anti tumor activity in GBM models. Crocetin treatment exhibits a greater effect than RT alone and similar effects to TMZ suggesting that is possible to combine crocetin with RT. This therapeutic approach could improve the antitumor effect of RT and reduce side effects and toxicity. So, we aimed to continue the study of crocetin’s efficacy in glioma disease, focusing our attention on the radio-sensitizing properties of this natural compound.

## Figures and Tables

**Figure 1 ijms-21-00423-f001:**
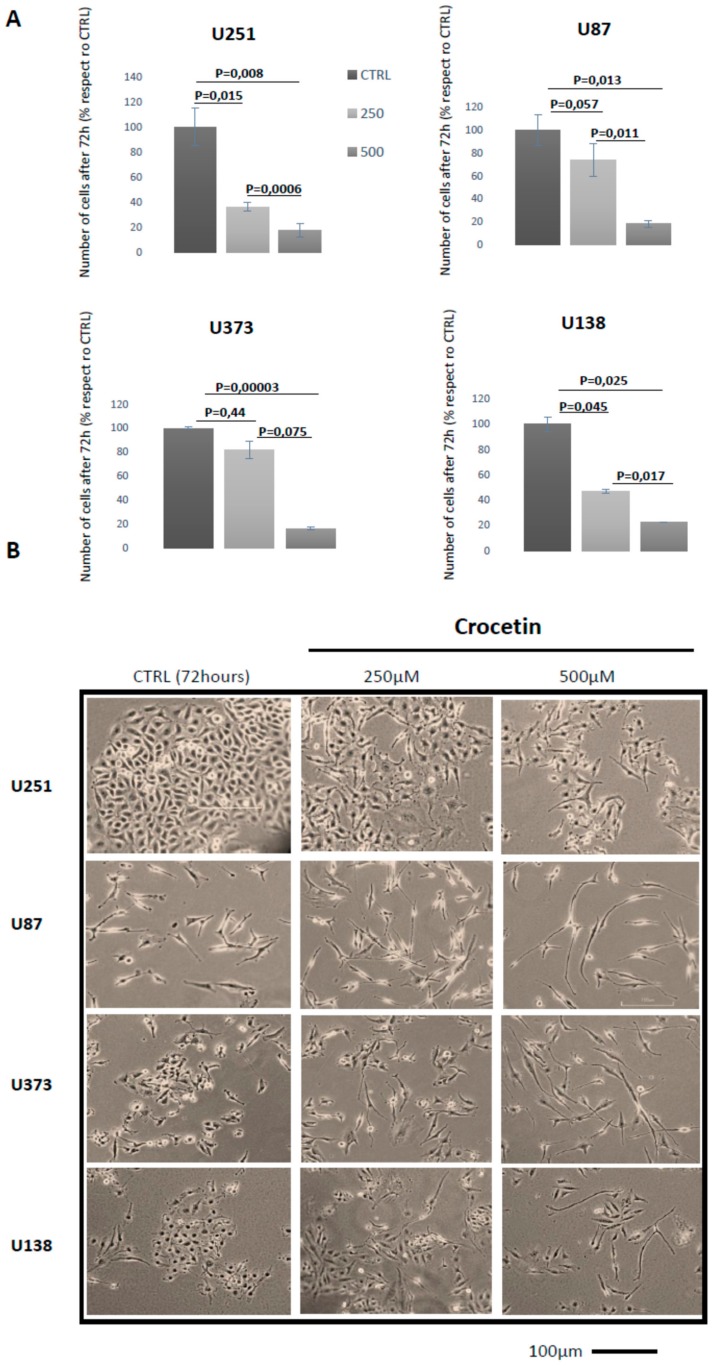
Crocetin (CCT) reduces proliferation and induces morphology changes in glioma cells. (**A**) Cell counts in U251, U87MG, U373, and U138 glioblastoma (GBM) cell lines performed at 72 h of treatment with 250 and 500 μM of crocetin. (**B**) Representative images of GBM cell lines in culture acquired at 40× magnification (bar corresponds to 100 μm).

**Figure 2 ijms-21-00423-f002:**
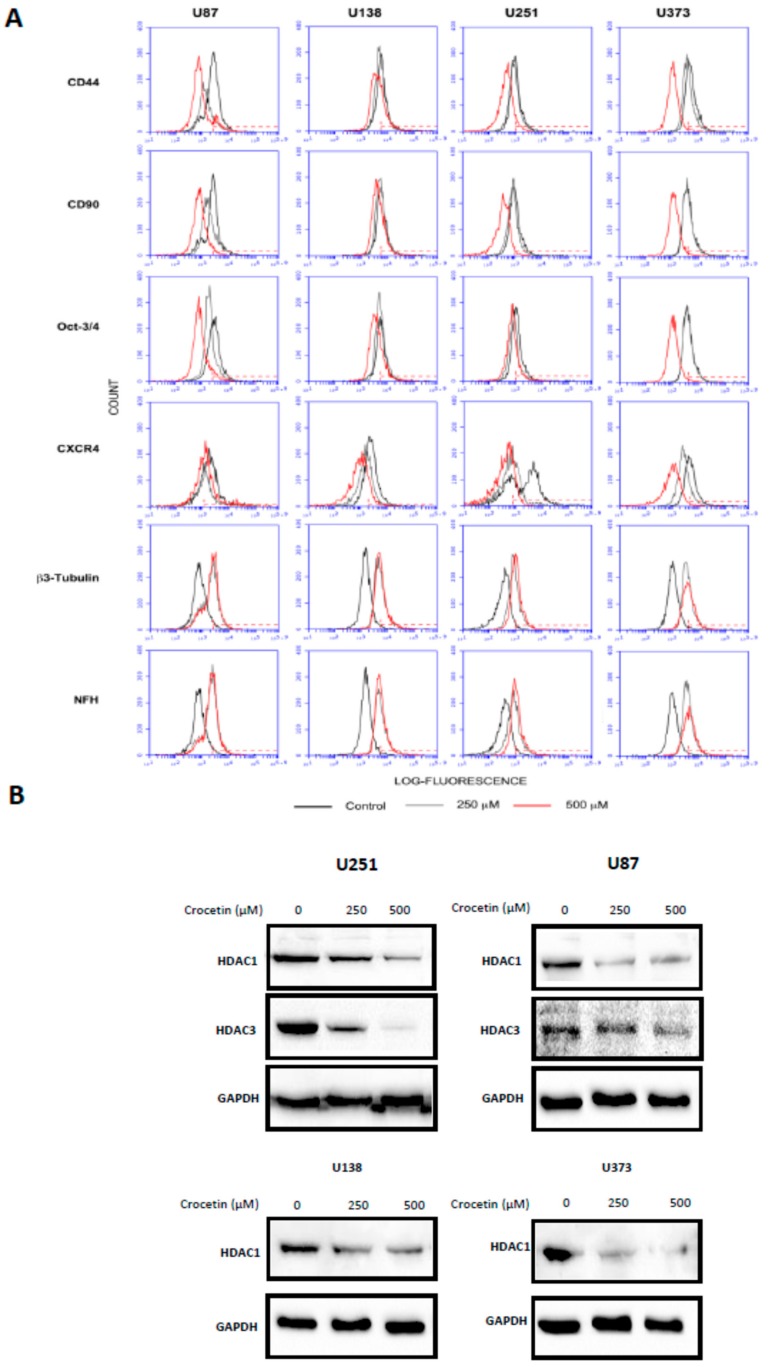
Crocetin reduces the levels of mesenchymal markers and induces an increase in neuronal ones in glioma cells, which could be related to histone deacetylase (HDAC) expression. (**A**) Representative Fluorescence-Activated Cell Sorter (FACS) histograms and (**B**) Western blotting analyses of HDAC1 and HDAC3 levels. Analyses was made at 72 h in cells after treatment with 250 and 500 μM of crocetin. Cell extract samples were loaded with 20 μg of protein per lane.

**Figure 3 ijms-21-00423-f003:**
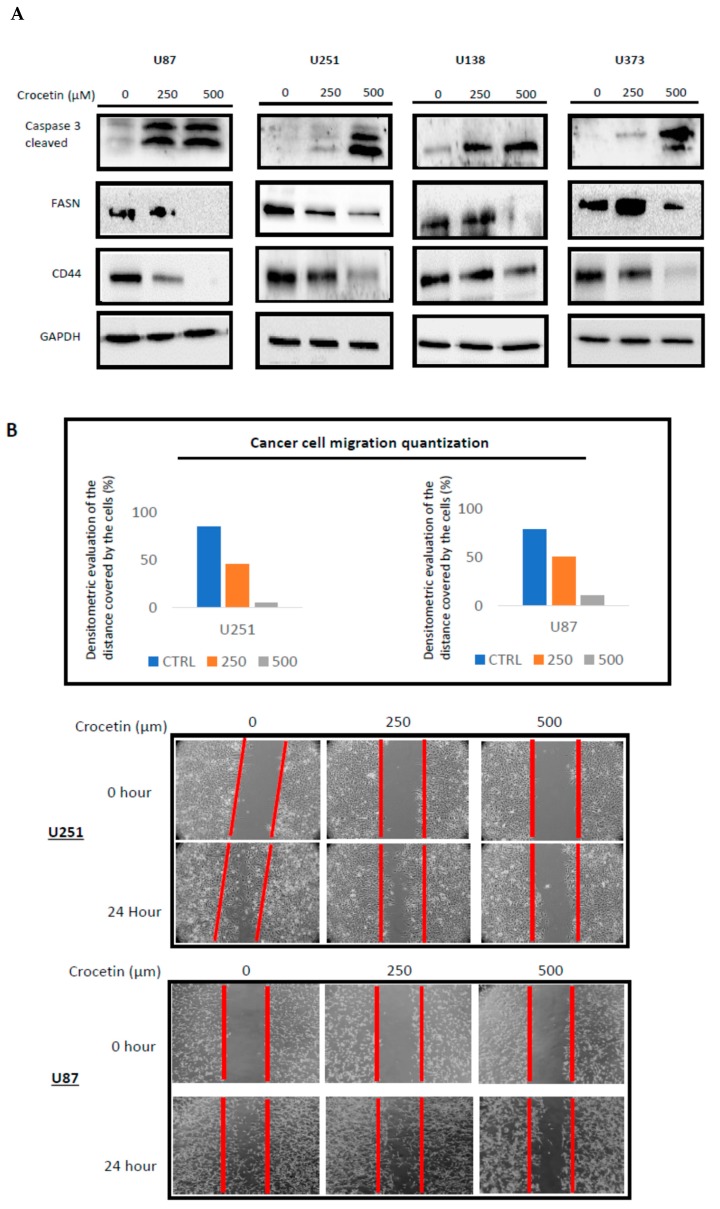
CCT inhibits the expression of FASN and CD44 proteins, thereby inducing cell apoptosis and reducing migratory capacity. (**A**) Western blot analyses of whole protein extracts harvested from treated (250 and 500 μM CCT) and untreated GBM cell lines for CD44, FASN, and activated caspase 3, for normalized versus GAPDH. Lanes were loaded with 20 μg of protein. (**B**) The wound-healing repair assay was performed on the U87MG and U251 cells. The images were acquired at T0 and T24 h. Quantization was performed by Image J software, as indicated in panel B.

**Figure 4 ijms-21-00423-f004:**
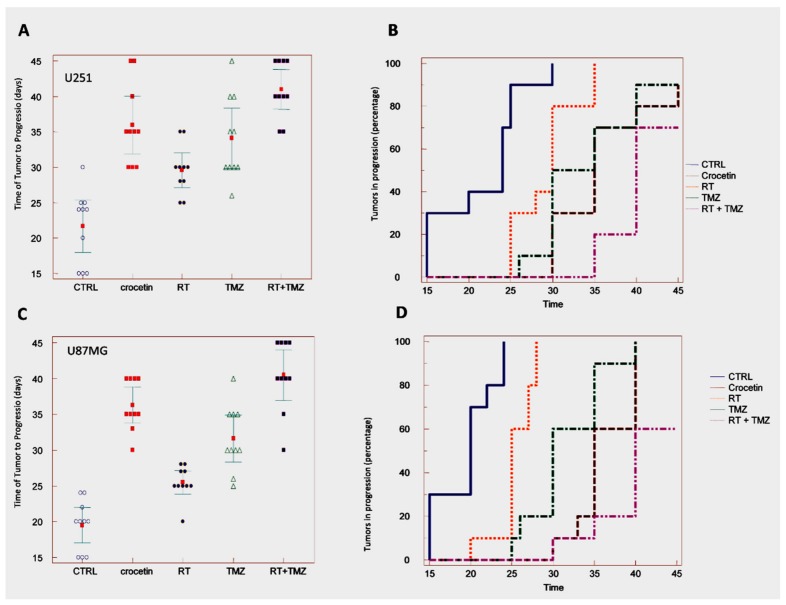
In vivo studies. Crocetin modifies tumor growth of GBM cells subcutaneously injected into female nu/nu mice (subcutaneous xenograft model). U251 and U87MG xenografts were treated with CCT (100 mg/kg/day per os), temozolomide (16 mg/kg for 5 consecutive days), RT (single dose of 4 Gy administered at day 5) and combination RT + TMZ (standard of care). (**A**) Time of tumor progression (TTP) calculated from the U251 xenograft. (**B**) Kaplan–Meier graphical analyses. (**C**) Time of tumor progression (TTP) calculated from the U87MG xenograft. (**D**) Kaplan–Meier graphical analyses.

**Figure 5 ijms-21-00423-f005:**
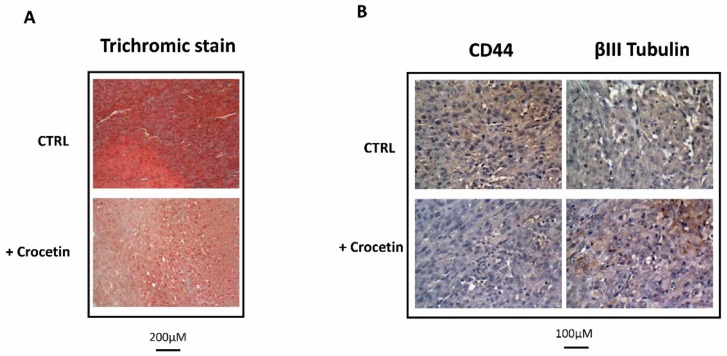
Histological and immunohistological analyses performed in tumors that were grown in the presence of a vehicle or CCT. (**A**) Trichromic staining performed at 20× magnification (bar represents 200 μm) and (**B**) CD44 and βIII tubulin expression on images harvested at 40× magnification. Bar represents 100 μm.

**Figure 6 ijms-21-00423-f006:**
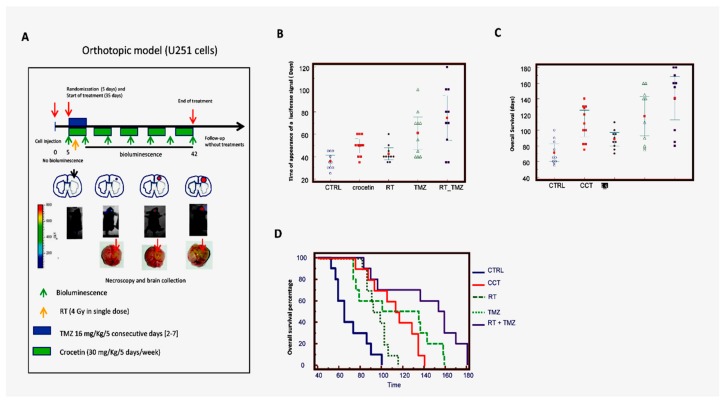
Crocetin increased disease-free and overall survival in orthotopic intra-brain tumors as determined by using a differentiated Luc-U251MG cell model. Luciferase-tagged U251 cells (Luc-U251) were orthotopically injected into the brains of CD1 mice, as described. After 5 days (when no tumors were organized into the brain) the animals were randomized and treated with CCT (100 mg/kg/day per os), temozolomide (16 mg/kg for 5 consecutive days), RT (single dose of 4 Gy administered at day 5), and a combination of RT + TMZ (standard of care). (**A**) Schema of treatments and performed analyses. (**B**) Disease-free survival graph representation. (**C**) Overall survival data: graphical representation. (**D**) Kaplan–Meier graphical analyses.

**Table 1 ijms-21-00423-t001:** Relative quantifications of the mesenchymal and neuronal markers in GBM cell lines as shown in Figure 2A.

% U87	CD44	CD90	OCT3/4	CXCR4	β3 Tubulin.	NFH
Controls (CTRL)	48.2	44.1	57.3	74.7	3.5	3.4
CCT 250 μM	16.6	24.7	21.0	55.8	37.5	39.1
CCT 500 μM	11.6	3.5	6.2	56.1	42.3	40.3
% U138						
Control	53.4	54	50.5	69.8	1.5	1.5
CCT 250 μM	43.4	46.5	46.9	35.5	40.9	42
CCT 500 μM	30.1	30.5	22.8	15	44.1	44.9
% U251						
Control	42.1	48.5	53.2	61.6	2.0	2.9
CCT 250 μM	32.7	27.9	27.2	34.5	35.6	32.3
CCT 500 μM	6.4	2.3	27.9	12.5	52.9	49.7
% U373						
ControlL	72	50.3	48.2	50.3	1.9	1.7
CCT 250 μM	47.8	48.6	47	14.9	42.6	47.3
CCT 500 μM	1.8	2.2	1.8	1.2	50.5	58.6

**Table 2 ijms-21-00423-t002:** Summarized statistical data from the TTP analysis, as seen in Figure 4A,C.

**U251**	**Mean ± SD**	**Statistics**
**Control**	21.7 ± 5.2	
**Crocetin**	36.0 ± 5.7	*p* < 0.0001 vs. CTRL
**RT**	29.6 ± 3.4	*p* = 0.0010 vs. CTRL *p* = 0.0081 vs. CCT
**TMZ**	34.1 ± 6.0	*p* < 0.0001 vs. CTRL *p* = 0.4759 vs. CCT (Not Significant, NS)
**RT + TMZ**	41.0 ± 3.9	*p* < 0.0001 vs. CTRL *p* = 0.0361 vs. CCT
**U87MG**	**Mean ± SD**	**Statistics**
**Control**	19.5 ± 3.5	
**Crocetin**	36.3 ± 3.5	*p* < 0.0001 vs. CTRL
**RT**	25.5 ± 2.3	*p* = 0.0003 vs. CTRL *p* < 0.0001 vs. CCT
**TMZ**	31.6 ± 4.6	*p* < 0.0001 vs. CTRL *p* = 0.0201 vs. CCT
**RT + TMZ**	40.5 ± 5.0	*p* < 0.0001 vs. CTRL *p* = 0.0447 vs. CCT

**Table 3 ijms-21-00423-t003:** Summarized statistical data from Kaplan Meier curves of Figure 4B,D.

**U251**	**Hazard Ratio**	**CI 95%**	**Statistics**
**CTRL vs. CCT**	4.58	1.49 to 14.10	*p* < 0.0001
**CTRL vs. RT**	3.08	1.11 to 8.54	*p* < 0.0011
**CTRL vs. TMZ**	4.05	1.36 to 12.10	*p* < 0.0001
**CTRL vs. RT + TMZ**	5.34	1.66 to 17.11	*p* < 0.0001
**RT vs. CCT**	2.75	1.01 to 7.45	*p* = 0.0044
**TMZ vs. CCT**	1.35	0.52 to 3.50	*p* = 0.4545 (NS)
CCT vs. RT + TMZ	2.36	0.83 to 6.37	*p* = 0.0465
**U87MG**	**Hazard Ratio**	**CI 95%**	**statistics**
**CTRL vs. CCT**	4.69	1.46 to 13.95	*p* < 0.0001
**CTRL vs. RT**	2.79	1.19 to 8.15	*p* = 0.0004
**CTRL vs. TMZ**	4.39	1.42 to 13.55	*p* < 0.0001
**CTRL vs. RT + TMZ**	5.52	1.73 to 17.62	*p* < 0.0001
**RT vs. CCT**	3.30	1.40 to 13.18	*p* < 0.0001
**TMZ vs. CCT**	1.67	0.67 to 4.12	*p* = 0.1059 (NS)
**CCT vs. RT + TMZ**	2.42	0.89 to 6.57	*p* = 0.0219

**Table 4 ijms-21-00423-t004:** Summarized statistical data of bioluminescence appearance times, as analyzed in Figure 6B.

U251	Mean ± SD	Statistics
**Control**	36.0 ± 7.4	
**Crocetin**	49.5 ± 9.9	*p* = 0.0019 vs. CTRL
**RT**	42.5 ± 7.5	*p* = 0.0472 vs. CTRL *p* < 0.0760 vs. CCT (NS)
**TMZ**	61.0 ± 20.4	*p* = 0.0038 vs. CTRL *p* = 0.1284 vs. CCT (NS)
**RT + TMZ**	64.5 ± 27.8	*p* < 0.0001 vs. CTRL *p* = 0.0205 vs. CCT

**Table 5 ijms-21-00423-t005:** Summarized statistical data of overall survival times, as analyzed in Figure 6C.

U251	Mean ± SD	Statistics
**Control**	71.3 ± 15.5	
**Crocetin**	108.7 ± 23.8	*p* = 0.0008 vs. CTRL
**RT**	88.5 ± 12.0	*p* = 0.0129 vs. CTRL *p* = 0.0324 vs. CCT
**TMZ**	117.8 ± 34.9	*p* = 0.0023 vs. CTRL *p* = 0.5055 vs. CCT (NS)
**RT + TMZ**	141.0 ± 38.5	*p* = 0.0002 vs. CTRLP = 0.0394 vs. CCT

**Table 6 ijms-21-00423-t006:** Summarized statistical data from the Kaplan Meier curves presented in Figure 6D.

U251	Hazard Ratio	CI 95%	Statistics
**CTRL vs. CCT**	CTRL vs. CCT	3.97	1.69 to 10.55
**CTRL vs. RT**	CTRL vs. RT	2.26	0.84 to 6.85
**CTRL vs. TMZ**	RT vs. CCT	2.67	0.93 to 8.49
**CTRL vs. RT + TMZ**	CTRL vs. TMZ	3.35	1.18 to 9.51
**RT vs. CCT**	CCT vs. TMZ	1.71	0.69 to 4.24
**TMZ vs. CCT**	RT + TMZ vs. CTRL	4.63	2.12 to 12.87
CCT vs. RT + TMZ	CCT vs. RT + TMZ	2.47	0.99 to 6.87

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
