# Peer review of "Crocetin Extracted from Saffron Shows Antitumor Effects in Models of Human Glioblastoma"

_ijms, 2020, doi:10.3390/ijms21020423_

Round 1

Reviewer 1 Report

A brief summary

In this manuscript, Colapietro et al. indicated the potential of a natural compound crocetin for GBM treatment both in vitro and in vivo. They demonstrated that crocetin is able to inhibit glioma cells proliferation and migration and to change glioma cell differentiated states with modulating HDAC expression. By using xenograft mouse model, they showed that crocetin halts the GBM growth and increases disease-free and overall survival, suggesting that crocetin is of the potential therapeutic use for GBM patients.

Broad comments

Overall, this is a novel finding about the effects of crocetin in GBM. Most experiments are well designed and explained. However, there are some issues needed to be addressed. I am also surprised that there are so many obvious errors.

Major comments:

In Fig 1, the concentrations of crocetin are different in the figure from the legend. The bars of U251 are much smaller than the others. It would be better to make them the same size. The illustration in fig 2 is hard to see especially in the printed form. Why is the expression of HDAC3 not determined in the U138 and U373. The quantified data of fig 3b is needed. I do not think that it is right to show the tables in the figures (fig 2b, fig. 4, and 5B). Usually, tables are shown separately. There are two fig 5. Why? It looks like that Fig 1 is not consistent with Fig 3. In Fig 1, crocetin is shown to inhibit glioma cells proliferation. However, in Fig 3, there is no obvious effects of crocetin on cell proliferation. Some labels are not accurate. For example, in some figures, only time is shown. What does time means? Hour, Day, or Week? There are two animal experiments in the method part. Why?

Minor comments: there are too many errors.

Line 50; Line 89; Line 106; Line 115; Line 237: reference is needed here. Line 325: reference is needed here.

Author Response

REVIEWER 1 

Question 1. In Fig 1, the concentrations of crocetin are different in the figure from the legend. The bars of U251 are much smaller than the others. It would be better to make them the same size.

This was fixed

Question 2. The illustration in fig 2 is hard to see especially in the printed form.

Reply: we have increased the quality of the image in figure 2.

Question 3. Why is the expression of HDAC3 not determined in the U138 and U373.

Reply: Our results indicated that HDAC3 expression in U138 and U373 cells didn’t change when crocetin treatment was administered. This information was added in the text. So, we decided to not show the bands relative to HDAC3 protein in the cells in which the protein resulted not modulated (data not shown).

Question 4. The quantified data of fig 3b is needed.

This was fixed and quantization was added in the figure.

Question 5. I do not think that it is right to show the tables in the figures (fig 2b, fig. 4, and 5B). Usually, tables are shown separately.

The tables previously shown in figures 2, 4 and 5 are split as tables I-VI accordingly.

Question 6. There are two fig 5. Why?

We thank the reviewer to have noticed this mistake. This was fixed in the text as figure 5 and figure 6.

Question 7. It looks like that Fig 1 is not consistent with Fig 3. In Fig 1, crocetin is shown to inhibit glioma cells proliferation. However, in Fig 3, there is no obvious effects of crocetin on cell proliferation.

Reply: in figure 1 and figure 3 we showed the results of two different experiments developed in different conditions. Indeed, in figure 3 we show the wound healing repair assay, an experiment commonly used for the study of migration of cells. In this case, the experiment requires that the cells in the plate reach 80-90% of confluence before to be treated and scratched, contrary to the proliferation analysis in which the cells are plated at a lower confluence. This is the first difference. The second difference is that for proliferation studies we maintain the cells in culture, with crocetin, for 72 hours; while in the wound healing repair assay we stopped the experiment and the observation at 24 hours of treatment. So, the two figures show a different cell behavior, in term of cell proliferation impairing, because they are in different experimental conditions and processed for the evaluation of two different cancer behaviour.

Question 8. Some labels are not accurate. For example, in some figures, only time is shown. What does time means? Hour, Day, or Week?

This was fixed

Question 9- There are two animal experiments in the method part. Why?

Reply: We used two animal model; (1) subcutaneous xenograft which allows to have data on tumor proliferation in the time and (2) orthotopic intra-brain xenografts which allow to have data on overall survival and state of health of animals. So, we used two separate section of M&M.

Minor comments: Line 50; 89; 106; 115; 237; 325 reference needed here.

Reply: This was fixed

Reviewer 2 Report

Thank you very much for sending me a manuscript entitled "Crocetin Extracted from Saffron Shows Antitumor Effects in Models of Human Glioblastoma" for review as a original manuscript. The authors try to explain the effect of crotecin on various glioma lines.

Currently, there is an unusual interest in compounds of natural origin with anti-cancer properties. The growing number of cases of different types of cancer forces work on alternative therapies or effective replacement of existing ones. It is known that plants are an extremely rich source of such compounds. Crocetin (8,8′-diapocarotenedioic acid), carotenoid derived from saffron indicates anti-cancer effects on many types of ctumors. 

The abstract and introduction are written correctly.
Line 99: in my opinion, more work should be cited describing the effect of various natural compounds on GBM. Only one work is cited here (22). similarly line 101.
Line 100: "saffron extract (SE) called crocetin (CCT)" it is not clear.
At the end of the introduction, the main goal of this work is not clearly described.
Materials and Methods:
1. There is no information from what amount of plant material crocetin was isolated.
Western blot:
2. Is it an original method or was it based on literature data. If based on other publications, please provide a reference.
3. Why did the authors choose the concentration of 250 and 500 µM crocetin for experiments? Is it based on literature data?
4.Histochemical and Immunohistochemical Analyzes:
Was the methodology developed by the authors or based on literature? Please provide references.
5. Why were proliferation reduction and morphological changes examined after 72 hours? Did you notice any changes after 24 and 48 hours?
Figure 3. In my opinion, gene expression should also be presented in graphs.
Line 325: if the dose of CCT was selected based on literature data, please provide them.

Author Response

REVIEWER 2

Question 1: Line 99: in my opinion, more work should be cited describing the effect of various natural compounds on GBM. Only one work is cited here (22). similarly line 101.

Reply: We added a couple of report as references 23, 24

Question 2: Line 100: "saffron extract (SE) called crocetin (CCT)" it is not clear.

statement was changed in: Saffron extract compound named crocetin

Question 3: At the end of the introduction, the main goal of this work is not clearly described.

Reply: This was fixed.

Question 4 There is no information from what amount of plant material crocetin was isolated.

Reply: Thank you for this question: we verified our records and found that the amount of crocetin extracted from dried stigmas was about 8-10% in weight as indicated in the new MM section.

Question 5. Western blotting reference.

Reply: For about western blot performing we have followed the protocol provided by the manufactures. We have better elucidate, after your review, the procedure of the technique in the materials and method.

Question 6. Why did the authors choose the concentration of 250 and 500 µM crocetin for experiments? Is it based on literature data?

Reply: In order to test the antiproliferative potential of crocetin in GBM cells, we saw that previous studies conducted in different cancer cell models, used a dose of crocetin near to 1 mM. so, we have tested a range of doses of crocetin between 0,1 and 1 mM, and we identified the doses of 250 and 500 microM as biological effective and not toxic for GBM cells.

Question 7. Histochemical and Immunohistochemical Analyzes:

Was the methodology developed by the authors or based on literature? Please provide references.

Reply: we used "standard protocols" for IHC staining and immuno-histochemistry analyses.

We added in the M&M the statement: For Histochemical and Immunohistochemical analysis we have performed the protocols provided by the manufactures.

Question 8. Why were proliferation reduction and morphological changes examined after 72 hours? Did you notice any changes after 24 and 48 hours?

Reply: Thank you for this question. We effectively appreciated a gradual inhibition of GBM cell proliferation and morphological changes at 24 and 48h yet, with crocetin treatment. However, we obtained the best results at 72h of treatment, with no toxic effect for GBM cells.

Question 9: Figure 3. In my opinion, gene expression should also be presented in graphs.

Reply: Thank you for the suggest. The analyses of gene expression (i.e. microRNA Profiling) will be one of future aims to characterize the effects of CCT in GBM. However, in the current work we focused on the evaluation only ofprotein expression of some important players of cell death and tumorigenesis in GBM (i.e. cleaved caspase 3, FASN, and CD44).

Question 10., Line 325: if the dose of CCT was selected based on literature data, please provide them.

Reply: The dose of crocetin for the in vivo treatment was selected based on the following reference:

this dose was previously considered in the our report [see reference 28]. However, a recent report considered that no toxicity of crocetin was find in mice treated up to 3.000 mg/Kg [see:ref 29]

Suparmi S, de Haan L, Spenkelink A, Louisse J, Beekmann K, Rietjens IMCM. Combining in vitro data and physiologically based kinetic modeling facilitated reverse dosimetry to define in vivo dose-response curves for bixin- and crocetin-induced activation of PPARγ in humans. Mol Nutr Food Res. 2019 Dec 17:e1900880. doi: 10.1002/mnfr.201900880

Round 2

Reviewer 1 Report

all my concerns have been well addressed by the authors. And I have no further comments and would like to suggest you consider to accept this MS.

Reviewer 2 Report

The authors have corrected the manuscript in accordance with my suggestions.